# Baseline Calibration Scheme Embedded in Single-Slope ADC for Gas Sensor Applications

Jang-Su Hyeon and Hyeon-June Kim *

Department of Semiconductor Engineering, Seoul National University of Science and Technology, Seoul 01811, Republic of Korea; hjs95111@seoultech.ac.kr
* Correspondence: hyeonjunekkim@seoultech.ac.kr

**Abstract:** This paper introduces a single-slope analog-to-digital converter (SS ADC) with an embedded digital baseline calibration scheme designed to improve the accuracy and reliability of gas sensor measurements. The proposed SS ADC effectively leverages an up/down counter mechanism to ensure stable signal extraction from gas sensors, despite variations in the baseline distribution. The proposed SS ADC initiates with a down counting operation to capture the initial output value of the gas sensor, which, after A/D conversion, is stored as a reference point for future readings. Subsequent gas sensor output values are derived by performing an up counting operation from this baseline reference. This approach allows for real-time correction of the baseline during the SS A/D conversion process, obviating the need for complex post-processing and baseline correction algorithms. The proposed SS ADC with the baseline calibration scheme was designed using a 0.18 μm standard CMOS process to confirm its feasibility. It demonstrated a signal-to-noise and distortion ratio (SNDR) of 57.56 dB and a spurious-free dynamic range (SFDR) of 59.02 dB, resulting in an effective number of bits (ENOB) of 9.27 bits in the post-simulation level. The proposed SS ADC has a total power consumption of 1.649 mW. This work offers an efficient solution to the baseline distribution problem in gas sensors, facilitating more reliable and accurate gas detection systems.

**Keywords:** gas sensor readout; baseline calibration scheme; single-slope analog-to-digital converter (SS ADC); up/down counter; on-chip self-calibration





## 1. Introduction

Recently, the marketability of gas sensor systems is rapidly expanding, driven by increasing demands across a broad spectrum of industries including environmental monitoring, industrial safety, healthcare, automotive, and smart buildings [1–3]. The critical role of gas sensors in identifying hazardous gases, tracking air quality, and safeguarding workplace safety underscores the pressing need for ongoing technological advancements in this domain [4]. Among the various types of gas sensors, semiconductor sensors stand out due to their distinct characteristics and advantages [5–8]. Semiconductor gas sensors, often based on metal oxide semiconductors, offer high sensitivity to a wide range of gases at relatively low costs [9,10]. These sensors operate on the principle that the resistance of the semiconductor material changes in the presence of target gases, enabling the detection of gas concentrations. The advantages of semiconductor sensors include their compact size, low power consumption, and ability to operate at room temperature, making them ideal for portable and fixed monitoring systems [11].

However, the baseline dispersion of gas sensors poses a significant challenge, including semiconductor types [12–14]. Baseline dispersion refers to the variability or drift in the sensor's baseline signal over time, which can occur due to changes in ambient conditions, the aging of the sensor, or exposure to background gases. This variability complicates the extraction of accurate and reliable signals from gas sensors, as it can mask the presence of target gases or lead to false readings. From the perspective of readout systems

designed to extract signals from gas sensors, addressing baseline dispersion is crucial for maintaining the precision and accuracy of gas measurements. That is, the difficulty lies in distinguishing between changes in sensor signals caused by target gas concentrations and those resulting from baseline dispersion. Addressing the issue of baseline dispersion in gas sensors is a critical challenge that can be mitigated through sophisticated post-digital processing techniques [15,16]. However, it is important to acknowledge that these solutions necessitate the use of complex processing algorithms and additional computation resources. This requirement may, in turn, impact the technological competitiveness of gas sensor systems, because the need for resource-intensive algorithms could potentially slow down the processing time or increase the cost of the sensor systems.

Meanwhile, by focusing on the readout integrated circuit (ROIC), another crucial component in the gas sensor system, we can uncover potential solutions to significantly alleviate the physical limitations inherent in the gas sensor itself [12,17,18]. Specifically, when it comes to the baseline dispersion problem of gas sensors, by integrating advanced correction techniques and algorithms within the ROIC, it is possible to dynamically adjust and compensate for the baseline in real time, enhancing the sensor's accuracy and reliability. This approach not only streamlines the post-calibration process but also mitigates the need for extensive post-processing, presenting a more efficient and effective strategy for overcoming challenges in gas sensor technology.

In this paper, we present an efficient baseline digital calibration scheme embedded in a single-slope (SS) ADC [12,19] for gas sensor readout. The proposed SS ADC enables stable gas signal extraction from gas sensors, unaffected by baseline distribution variations. This method involves initially down counting the output value of the gas sensor at the start of the measurement. After the analog-to-digital (A/D) conversion, this initial value is stored and serves as a reference point for subsequent measurements. The output value of the gas sensor is then extracted by up counting from this stored value for each new measurement. Through this SS A/D conversion process, the baseline value of the gas sensor is effectively corrected in real time. This approach circumvents the need for complex post-processing and baseline correction algorithms, allowing for the precise extraction of changes in the gas sensor's output signal without the requirement of manually removing or adjusting the baseline drift.

The remainder of this paper is organized as follows: Section 2 examines the characteristics of the baseline variation. Section 3 describes the proposed SS ADC with the baseline digital calibration scheme. The simulation results of the proposed scheme are discussed in Section 4. Finally, the conclusions are presented in Section 5.

## 2. Characteristics of Baseline Variation

Figure 1 shows the baseline distribution across different gas sensor samples. The gas sensor outputs encapsulated by the red box show the output results stabilizing after the introduction of air, representing a baseline variation situation. Gas sensors [20,21] are characterized by their resistance change ($R_S$) in response to gas concentrations, a fundamental property that allows for the detection of gas presence and concentration levels. To convert this resistance change into a measurable output, a load resistance ($R_L$) is utilized for resistance-to-voltage (R-V) conversion, using a supply voltage ($V_{DDS}$) to transform the gas concentration into a voltage signal ($V_O$) [22]. However, a significant challenge arises due to the intrinsic material characteristic variability among gas sensors. This variability leads to differing baseline resistances, resulting in disparate output values ($V_O$) for the same gas concentration across sensors. Such baseline variation complicates the accurate extraction of gas concentration information, as a standard R-V conversion circuit may not account for these discrepancies effectively. In light of this, accurate gas concentration determination becomes challenging with conventional R-V conversion circuits alone, necessitating additional post-processing digital signal processing (DSP) after the initial signal extraction. This requirement for complex DSP not only increases the processing burden but also impacts the technological competitiveness of gas sensor systems by adding layers of complexity

and potential points of failure. To address the challenges of accuracy and reliability in gas concentration measurements, efforts to develop embedded baseline calibration methods within readout circuit designs are necessary. By enabling gas sensor readout circuits to adapt to baseline variability during signal extraction, the dependency on post-processing DSP can be reduced, thereby improving the performance and competitiveness of gas sensor systems. Based on these motivations, this paper introduces a simple yet powerful baseline calibration technique embedded within the readout ADC design, along with verification results demonstrating its effectiveness.

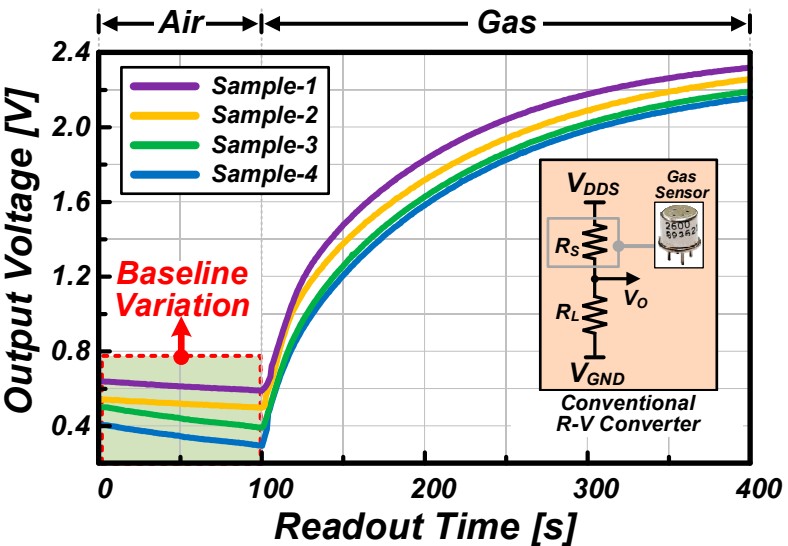

**Figure 1.** Characteristic of baseline variation inherent in gas sensor.

## 3. Proposed SS ADC with Baseline Calibration Scheme

### 3.1. Typical SS ADC for Gas Sensor Readout

Figure 2a provides a simplified block diagram of a conventional single-slope analog-to-digital converter (SS ADC), which is particularly advantageous for integration with gas sensors due to its simplicity and effective linear performance [12,23]. It is composed of a comparator that contrasts the input signal with a reference signal, a ramp generator responsible for producing the reference signal for A/D conversion, an N-bit counter that counts the digital code until the comparator's output trigger, and the digital control logic that manages the overall A/D conversion process. Figure 2b illustrates an operational timing diagram of the conventional SS ADC. When the $\varnothing_{IN}$ is switched on, the input voltage ($V_{IN}$) is sampled onto the sampling capacitor ($C_{SH}$) and subsequently directed to the negative node of the comparator ($V_{INN}$). With the activation of the $\varnothing_{CEN}$, the ramp generator begins to advance the A/D reference voltage ($\Delta V_{RAMP}$), which propels the positive node ($V_{INP}$). The counter begins to increment the digital code step by step as soon as the ramp voltage ($V_{INP}$) starts to rise. It continues this process until $V_{INP}$ rises below the sampled input voltage at $V_{INN}$. At this moment, the comparator generates an output signal ($V_{OUT}$) which halts the counter's progression. The digital code value registered at this instant is then used to represent the A/D conversion result for the input voltage ($V_{IN}$).

The SS ADC architecture is notably less complex than other ADC architectures, which is especially beneficial when employed in gas sensor readout ADCs. This simplicity ensures that the slow response times characteristic of gas sensors do not adversely affect the ADC's performance, thus maintaining real-time monitoring capabilities. Additionally, the single-slope configuration reduces the overall complexity of the ADC design, potentially lowering manufacturing costs and facilitating scalability within extensive sensor networks. The SS ADC's compatibility with the gas sensors' slow dynamics makes it an ideal choice for accurate and reliable gas detection required in various applications.

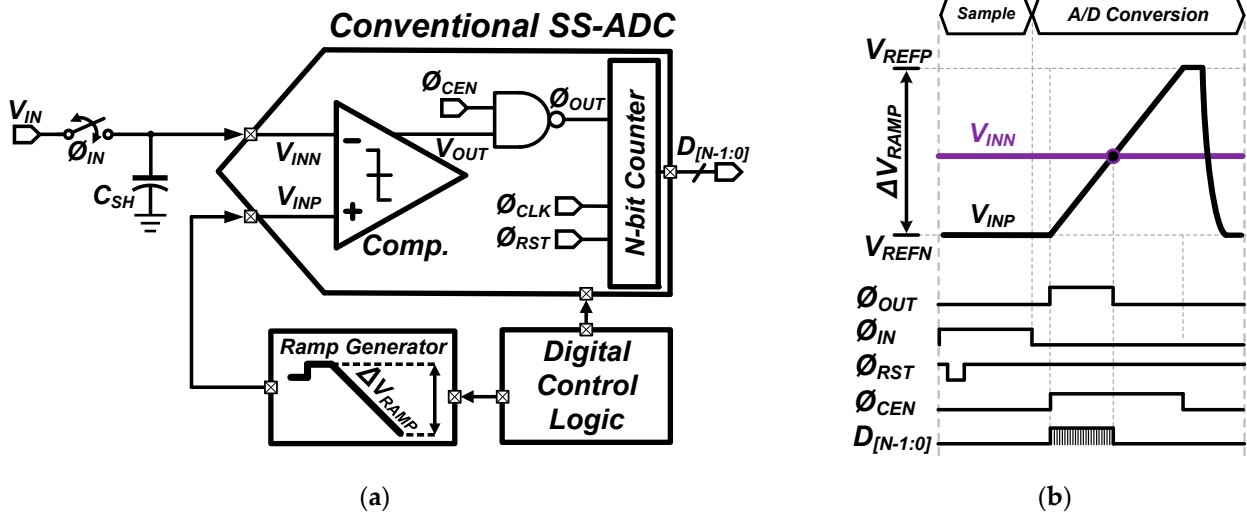

**Figure 2.** (**a**) Simplified block diagram of conventional SS ADC and (**b**) its operational timing diagram.

Figure 3 shows a simplified conventional N-bit binary up counter, which is typically used in SS ADCs [24]. This counter is composed of toggle flip-flops (T-FFs), with the output Q of each flip-flop representing a digital output bit of the counter. The first T-FF in the sequence is known as the least significant bit (LSB) T-FF, which receives the clock pulse ($\varnothing_{CLK}$) and is the first to toggle. Each T-FF in the chain represents an increasing bit significance, forming the binary count as the circuit operates. When the flip-flop toggles, it signifies a binary '1' in that bit position, and when it is reset, it represents a binary '0'. The counting sequence is initiated by the $\varnothing_{CLK}$ that is applied to the LSB T-FF. Subsequent flip-flops receive the clock input from the preceding flip-flop's output (Q), thus creating a cascading count effect throughout the bit chain. The Q of each T-FF indicates an individual bit of the final binary count, with the collection of these outputs (D[0] to $D_{[N-1]}$) representing the digital count corresponding to the analog input signal. The asynchronous reset input ($\varnothing_{RST}$) allows for the entire counter to be reset instantaneously. When a 'Low' is applied to $\varnothing_{RST}$, all Qs of the T-FFs are driven to '0', effectively clearing the counter and setting it to its initial state. The counter's operation is controlled by a combined enable signal derived from the NAND gate that processes the counter enable signal ($\varnothing_{CEN}$) and the comparator's output signal ($V_{OUT}$). This enable signal allows the counter to proceed with counting only when $\varnothing_{CEN}$ is active, and $V_{OUT}$ is in a state indicating that the comparator has not yet detected the $V_{RAMP}$ reaching the $V_{IN}$. The counter stops counting when the comparator outputs the $V_{OUT}$, indicating that the $V_{RAMP}$ has reached the $V_{IN}$, thereby capturing the count at that precise moment as the digital equivalent of the analog input signal. This design ensures that the SS ADC can accurately and efficiently convert analog inputs into digital representations by counting in a binary sequence corresponding to the duration for which the reference voltage exceeds the input signal.

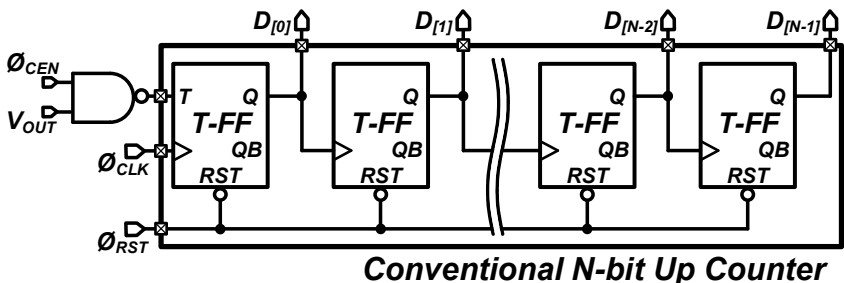

**Figure 3.** Simplified conventional N-bit up counter for SS A/D conversion.

### 3.2. Proposed Baseline Calibration Scheme

The slow response characteristics of gas sensors are important factors to consider when implementing an effective baseline calibration scheme. To compensate for baseline variations that may vary with environmental variables such as temperature, humidity, etc., it is suitable to digitally store baseline information. Effectively, if the change in gas sensor output relative to this stored baseline information can be quantified through A/D conversion, it would facilitate the implementation of the baseline calibration scheme.

Figure 4 shows a simplified schematic of the proposed SS A/D counter with an embedded baseline calibration scheme specifically designed to address the slow response characteristics of gas sensors. The proposed counter is based on a 10-bit up/down counter structure [25] to achieve a 10-bit A/D resolution, with an added most significant (MSB) bit serving as a sign bit ($D_{[SN]}$). An additional 11-bit baseline register is incorporated to update the baseline value of the gas sensor, with its input connected to the digital output bus $D_{[SN:0]}$ and its output linked to the multiplexers (MUXs) located at the counter's reset and preset inputs. The counter's operation range, defined by $D_{[N-1]}$:$D[0]$, spans from $-(2^{(N-1)} - 1)$ to $(2^{(N-1)} - 1)$, allowing for both positive and negative decimal values with the MSB indicating the sign. The counter can select between the Q and QB outputs from each stage through the MUX switches, controlled by the external signal $\varnothing_{U/D}$. When $\varnothing_{U/D}$ is 'HIGH', Q is selected for up counting, and when $\varnothing_{U/D}$ is 'LOW', QB is chosen for down counting. Moreover, when $\varnothing_{UPDE}$ is 'HIGH', the outputs of the register $D_{REG[SN:0]}$ and $\overline{D_{REG[SN:0]}}$ are applied to the T-FF's reset (RST) and preset (PRE), respectively. If $D_{REG[N]}$ is 'LOW', the T-FF's RST is activated, setting the corresponding digital output bit $D_{[N]}$'s Q to 0. Conversely, if $D_{REG[N]}$ is 'HIGH', the T-FF's PRE is activated, setting $D_{[N]}$'s Q to 1. This operation mechanism allows for dynamic updating of the baseline value in response to environmental changes, ensuring accurate gas sensor readings.

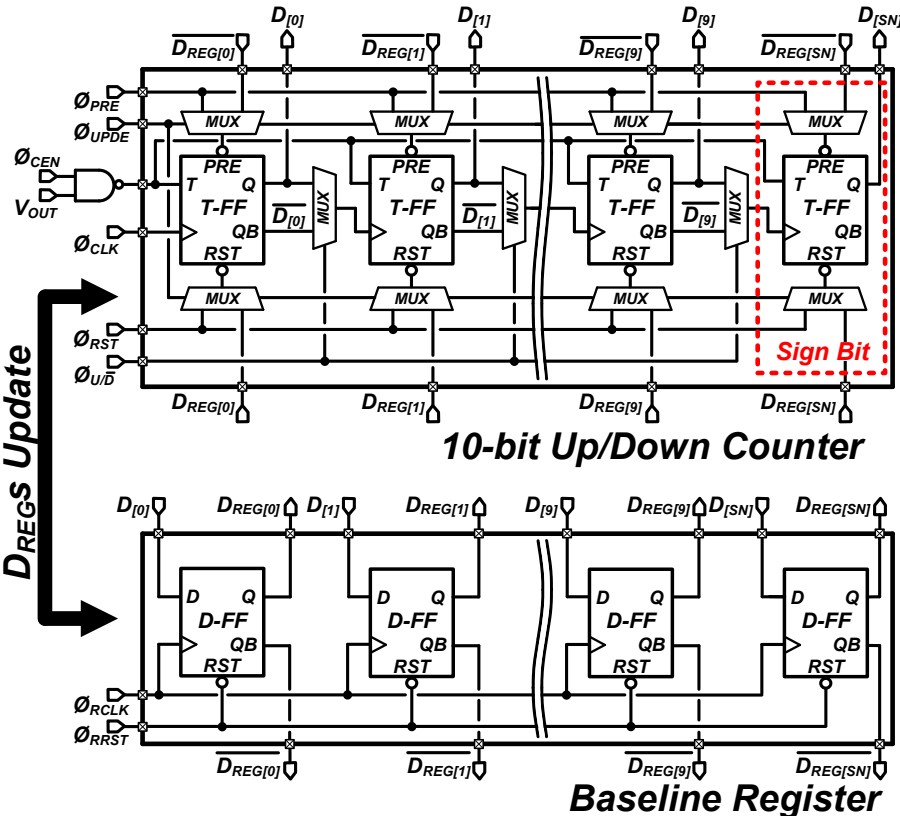

**Figure 4.** Simplified schematic of proposed SS A/D counter with baseline calibration scheme.

Figure 5 provides operational examples of the proposed counter structure with the baseline calibration scheme, illustrating cases when the sign bit ($D_{[SN]}$) is set to '0' and

'1'. The operation procedure is divided into two phases: the baseline readout phase for establishing the reference point and the signal readout phase for measuring the gas concentration. During the baseline readout phase, the initial value of the gas sensor, which serves as the baseline, is captured to commence SS A/D conversion. At this stage, with the $\varnothing_{U/D}$ signal set to low, the proposed counter conducts down counting until the comparator is triggered, obtaining the baseline value ($-D_{BL}$). This value is then stored in the baseline register as $D_{REG}$ when triggered by $\varnothing_{RCLK}$. In the signal readout phase, the $-D_{BL}$ value is updated as the initial value of the proposed counter each time $\varnothing_{UPDE}$ is triggered. With $\varnothing_{U/D}$ set to high, the counter performs up counting from the $-D_{BL}$, effectively extracting the output signal of the gas sensor. Since the counter starts from $-D_{BL}$, the gas sensor's baseline value is compensated for during the conversion process, and the SS A/D conversion result is represented as $D_{SIG} - D_{BL}$. Figure 5a represents the case where $D_{SIG} - D_{BL} \geq 0$, indicating that the absolute value of $D_{SIG}$ is greater than the initial $D_{BL}$. Here, the sign bit $D_{[SN]}$ is set to 'LOW', and the final output code has a positive sign with a magnitude of $D_{SIG} - D_{BL}$. Conversely, Figure 5b depicts the case where $D_{SIG} - D_{BL} < 0$, meaning the absolute value of $D_{SIG}$ is less than $D_{BL}$. In this instance, $D_{[SN]}$ is set to 'HIGH', and the final output code carries a negative sign with a magnitude of $D_{SIG} - D_{BL}$. Through this process, by extracting a signal that correlates with the initial baseline code of the gas sensor, the output $D_{SIG}$ effectively performs baseline digital calibration without the need for additional post-processing digital calibration. This scheme ensures that the gas sensor's output is accurately represented in relation to its baseline, thus facilitating a more streamlined and efficient signal processing operation.

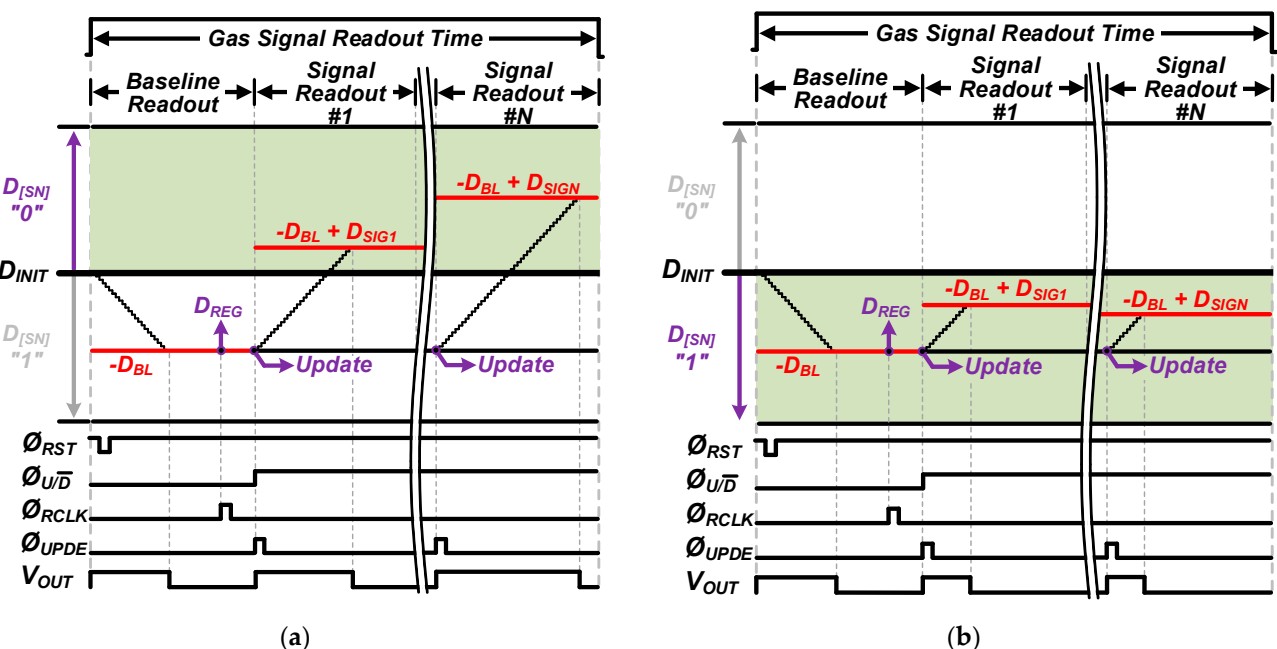

**Figure 5.** Operation examples of proposed counter with baseline calibration scheme: (**a**) case for $D_{[SN]}$ = '0' and (**b**) case for $D_{[SN]}$ = '1'.

Figure 6 shows the overall process of the proposed baseline calibration algorithm embedded in the SS ADC. The operational algorithm is bifurcated into two distinct cases: baseline readout and signal readout. For the initial A/D conversion, the algorithm carries out the A/D conversion for the baseline. It commences by resetting $D_{OUT}$ to zero. If $V_{IN}$ is greater than $V_{RAMP}$, the algorithm executes the down counting operation by decrementing $D_{OUT}$ by one LSB ($D_{OUT} = D_{OUT} - 1$). Concurrently, $V_{RAMP}$ is incremented by one LSB in a process that is synchronized with the counter clock, repeating this sequence until $V_{IN}$ becomes less than $V_{RAMP}$. At this moment, $D_{OUT}$ is stored in $D_{REG}$, which is then utilized as the baseline ($-D_{BL}$). If the operation does not consist of the initial A/D conversion,

$D_{OUT}$ is set to the stored $-D_{BL}$ value. When $V_{IN}$ exceeds $V_{RAMP}$, $D_{OUT}$ undergoes an up counting operation, increasing by one LSB. Once $V_{IN}$ becomes less than $V_{RAMP}$, the current value of $D_{OUT}$ represents the actual change in the gas sensor's signal and concludes the A/D conversion process. In the proposed baseline calibration algorithm, the down counting and up counting mechanisms are key points for dynamically adjusting the A/D conversion process based on $V_{IN}$, ensuring that the ADC output reflects real-time changes in gas concentration.

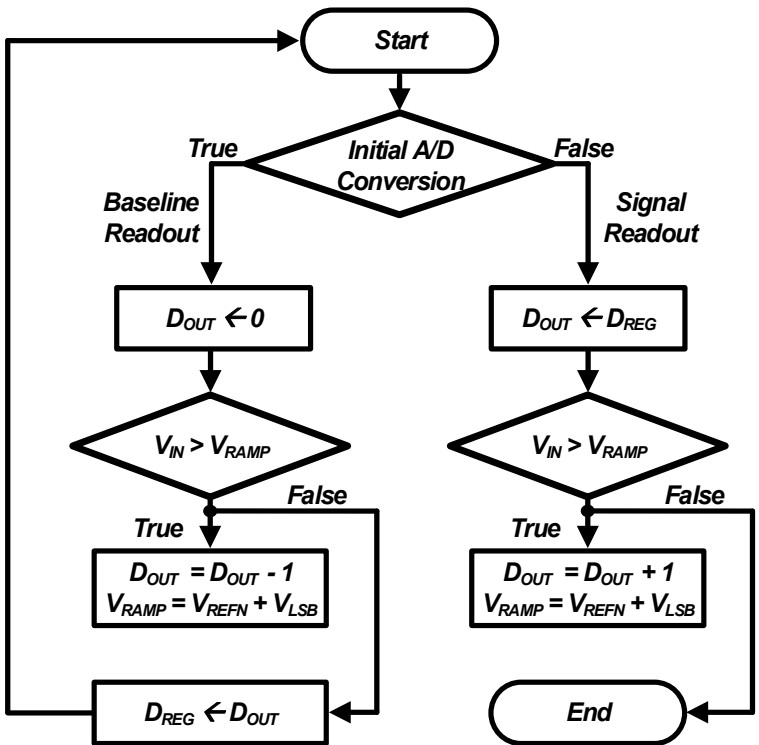

**Figure 6.** Operational algorithm for proposed SS ADC with baseline calibration scheme.

Figure 7 illustrates the effectiveness of the proposed baseline calibration scheme for gas sensor signal extraction when faced with baseline voltage variations due to environmental changes or sensor activation. This demonstrates how the proposed scheme accommodates $V_{BL}$ and extracts the actual change in the gas sensor signal, $\Delta V_{SIG}$. Initially, before gas exposure, the SS ADC converts $V_{BL}$ into $D_{BL}$, which is stored in the baseline register. This value is then used as the A/D initial value for subsequent signal extractions. It is assumed that the gas sensor is adequately activated before gas exposure. For instance, in Section 1 of Figure 7, even though the maximum sensor output is $V_{MAX1}$, the proposed SS ADC process extracts only the actual signal change, $\Delta V_{SIG1}$. Since $V_{IN}$ is greater than or equal to $V_{BL1}$, $D_{[SN]}$ is set to '0', resulting in positive output codes $D_{S1}$ and $D_{S2}$ for $\Delta V_{S1}$ and $\Delta V_{S2}$, respectively. In Section 2, $V_{IN}$ falls below $V_{BL1}$, hence $D_{[SN]}$ is set to '1', and the changes in $\Delta V_{S3}$ and $\Delta V_{S4}$ are converted into negative output codes $D_{S3}$ and $D_{S4}$. Section 3 addresses a case where the original baseline has shifted, and the system updates from $-D_{BL1}$ to $-D_{BL2}$ to capture $\Delta V_{SIG2}$. Similarly to Section 1, since $V_{IN}$ is greater than or equal to $V_{BL2}$, $D_{[SN]}$ remains as '0', and the changes in $\Delta V_{S5}$ and $\Delta V_{S6}$ are converted into positive output codes $D_{S5}$ and $D_{S6}$. Without applying the proposed technique, a complex post-processing step would be required to extract $\Delta V_{SIG}$ information from the A/D conversion data of $V_{MAX}$. However, with the application of the proposed method, $\Delta V_{SIG}$ values can be directly and effectively extracted, which reduces computational demands. By minimizing the need for additional processing, this method can potentially enhance the gas sensor system's overall performance and response time.

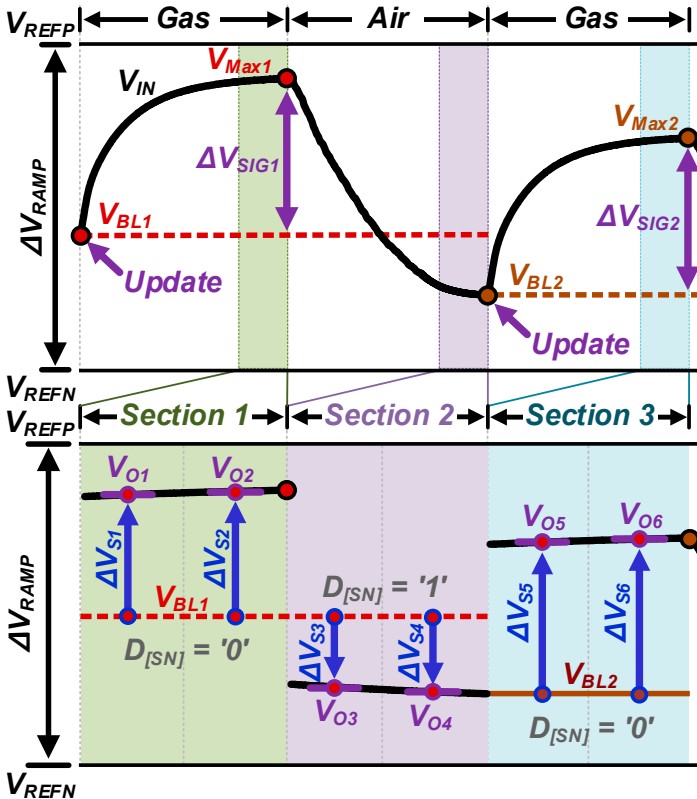

**Figure 7.** Explanation for gas sensor signal extraction using proposed baseline calibration scheme.

## 4. Simulation Results and Discussion

Figure 8 presents the post-simulation results of the proposed baseline calibration scheme, showing the cases when $D_{[SN]}$ is '0' in Figure 8a and '1' in Figure 8b. Here, the simulations were conducted using the TSMC 180 nm 1P6M CMOS process at a standard temperature of 300 K and at the typical process corner (tt). Furthermore, we incorporated transient noise ranging from 0.1 Hz to 500 kHz. $V_{IN}$ represents the output voltage of the gas sensor. The simulations were conducted by setting the initial $V_{IN}$ to different levels of $1.6V_{BL}$, $1.5V_{BL}$, and $1.4V_{BL}$, after which $V_{IN}$ was increased at a constant slope. In Figure 8a, where the gas is introduced at a voltage level higher than $V_{BL}$, $D_{[SN]}$ outputs '0', indicating a positive output code value. During the baseline readout section, the $\varnothing_{RST}$ signal triggers the reset of the SS ADC's counter initial value ($D_{OUT} = 0$). With the $\varnothing_{U/D}$ signal at 'LOW', the proposed counter performs down counting during SS A/D conversion, outputting codes '−631', '−561', and '−491' for $1.6V_{BL}$, $1.5V_{BL}$, and $1.4V_{BL}$, respectively. Subsequently, $D_{OUT}$ is stored as $D_{BL}$ in $D_{REG}$ upon triggering $\varnothing_{RCLK}$. From the signal readout section onwards, the $\varnothing_{U/D}$ signal switches to 'HIGH', and the counter engages in up counting during SS A/D conversion. When $\varnothing_{UPDE}$ is triggered, the counter's initial value is updated with the stored $D_{BL}$, and SS A/D conversion is performed based on this value. The simulation shows that regardless of the initial condition decreasing from $1.6V_{BL}$ to $1.4V_{BL}$, the counter's output, $D_{OUT}$, increases uniformly in response to the input's consistent slope. Similarly, in Figure 8b, under conditions where the input is introduced at a voltage level lower than $V_{BL}$, $D_{[SN]}$ outputs '1', resulting in negative output code values. It is observed that for different $V_{BL}$ levels, $D_{OUT}$ decreases uniformly in response to inputs that decrease with the same rate. This simulation demonstrates the effectiveness of the proposed calibration technique in maintaining consistent signal extraction across varying baseline levels, showcasing the robustness of the SS A/D conversion process within the gas sensor system. The ability to adjust the counter's initial value according to the stored $D_{BL}$ ensures accurate signal processing and calibration, affirming the potential of this method to reduce computational demands and enhance the overall performance and response

time of the gas sensor system. Furthermore, the proposed baseline calibration scheme is versatile and can be applied to a variety of gas sensor types. It is particularly beneficial for addressing issues of initial value dispersion that can occur even among sensors of the same type [26–28].

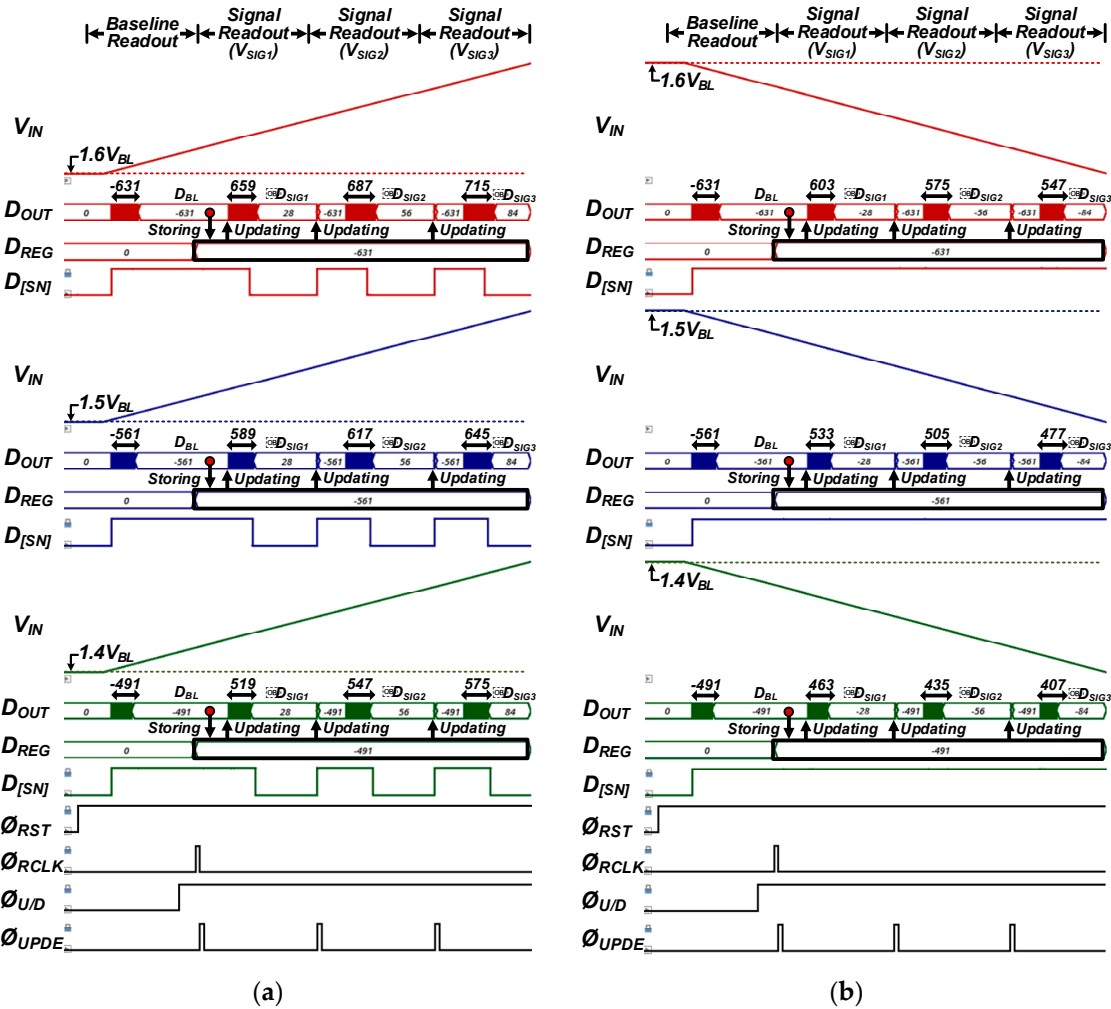

**Figure 8.** Simulation results of proposed baseline calibration scheme: (**a**) case for $D_{[SN]}$ = '0' and (**b**) case for $D_{[SN]}$ = '1'.

Figure 9 presents the fast Fourier transform (FFT) analysis results to evaluate the dynamic performance of the proposed SS ADC with the baseline calibration scheme. FFT is derived from post-simulation based on 512 samples of a sinusoidal input signal of 164.06 Hz ($F_{IN}$), in consideration of the gas sensor's slow transient response. The peak-to-peak voltage of the sine wave ($V_{IN}$) was set to 1.2 V, within the ADC's input range, and the ADC's sampling frequency ($F_S$) was simulated at 28 kHz. Here, the FFT output is displayed on a logarithmic scale along the *x*-axis, considering the low input frequency. Additionally, the post-simulation was conducted at room temperature (300 K), including transient noise with a frequency range from 0.1 Hz up to 500 kHz. As a result, the proposed SS ADC achieved a signal-to-noise and distortion ratio (SNDR) of 57.56 dB, resulting in an effective number of bits (ENOB) of 9.27 bits and a spurious-free dynamic range (SFDR) of 59.02 dB.

Figure 10 displays a post-simulation result for differential non-linearity (DNL) and integral non-linearity (INL) of the proposed SS ADC, sampled at an Fs of 28 kHz. The DNL characteristics range from a maximum of +0.259 LSB to a minimum of −0.263 LSB. Meanwhile, the INL characteristics are shown to range from a maximum of +0.868 LSB to a minimum of −0.122 LSB.

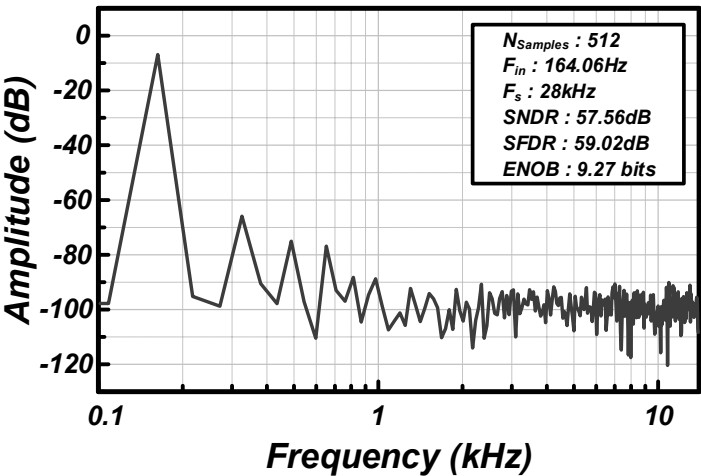

**Figure 9.** Fast Fourier transform analysis results of proposed SS ADC with baseline calibration scheme.

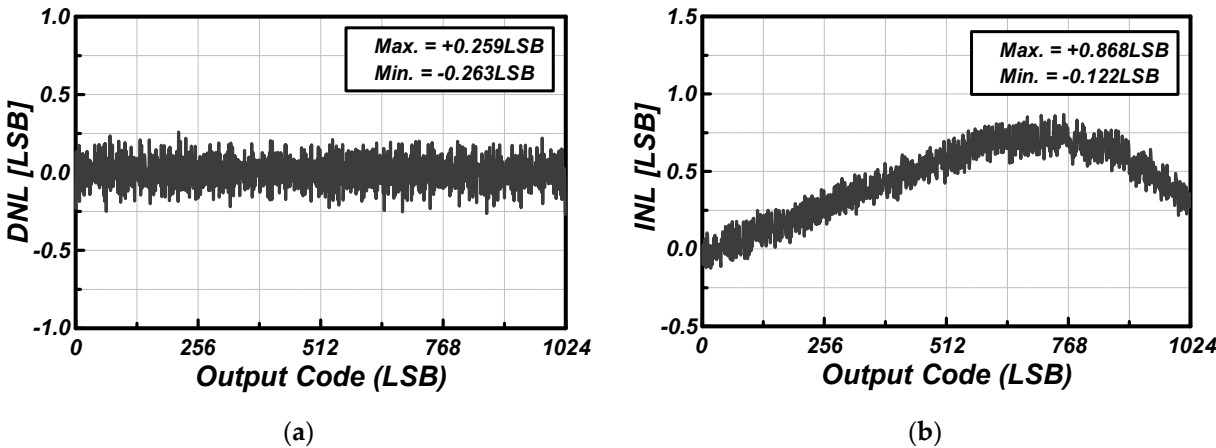

(**a**)                                                        (**b**)

**Figure 10.** (**a**) Differential non-linearity and (**b**) integral non-linearity.

Figure 11 illustrates the total power consumption of the proposed SS ADC. It comprises several components: the comparator, ramp generator, biasing circuit, and counter, resulting in a total power consumption of 1.649 mW. The counter, embedded with the proposed baseline calibration scheme, consumes a total of 39.42 µW. This figure represents an approximate 11% increase compared to the conventional counter structure, but it only constitutes about 0.24% of the total power consumption. Moreover, since the proposed SS ADC performs baseline calibration during the gas sensor signal extraction, it can actually be considered to offer a power-saving benefit when digital computing resources are taken into account.

Table 1 presents a comparison between the conventional counter and the proposed counter, detailing the differences in logic, the number of transistors, and area usage. Specifically, the proposed counter includes an additional T-FF for the MSB sign bit, and an 11-bit D-FF register for storing the baseline code. It also includes 31 MUX switches to facilitate the up/down counter function and to control the reset and preset of the T-FFs. This addition of logic results in an increase in the total number of transistors from 438 to 1222. However, when considering the overall chip area, the actual increase in area is only about 2%, which is negligible in the total chip area. This indicates that the proposed scheme can implement an effective gas sensor baseline calibration function with a minimal increase in area cost.

Figure 12 depicts the layout of the proposed SS ADC with the baseline calibration scheme. It is fabricated using a 180 nm standard CMOS process. The entire chip occupies an area of 1.4 mm by 1.5 mm. Within this chip, the area specifically allocated for the proposed counter measures 121 µm by 155 µm. Table 2 provides a performance summary.

Specifically, the performance of gas sensors is directly linked to the characteristics of the sensitive materials used, the heating process employed to enhance sensitivity, and the susceptibility to transient noise from external sources. These factors critically influence the reliability of the initial baseline conversion and, consequently, the overall system performance adopting the proposed baseline calibration scheme. Hence, there is a clear necessity for the development of robust algorithms capable of mitigating such challenges. Future research will be dedicated to exploring advanced algorithmic solutions that address these issues, aiming to enhance the resilience and accuracy of gas sensor systems.

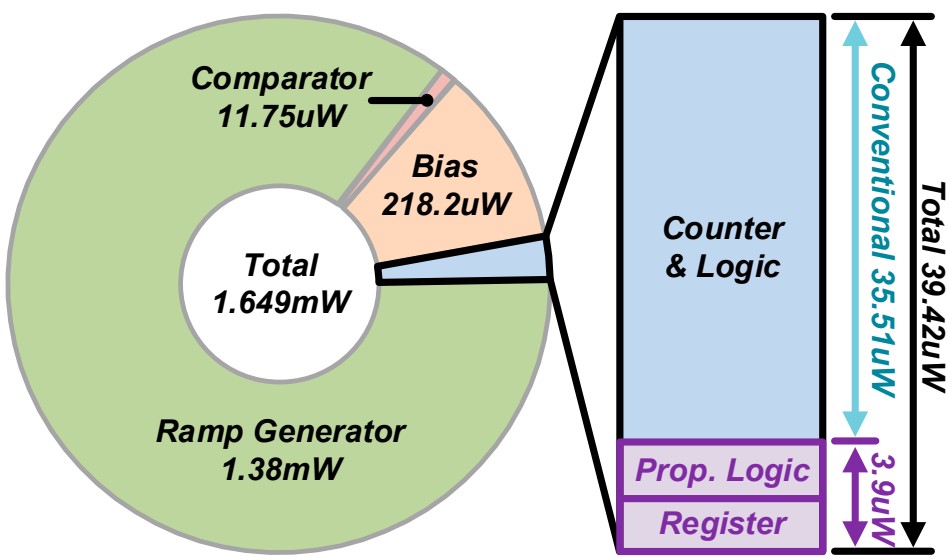

**Figure 11.** Power consumption of proposed SS ADC.

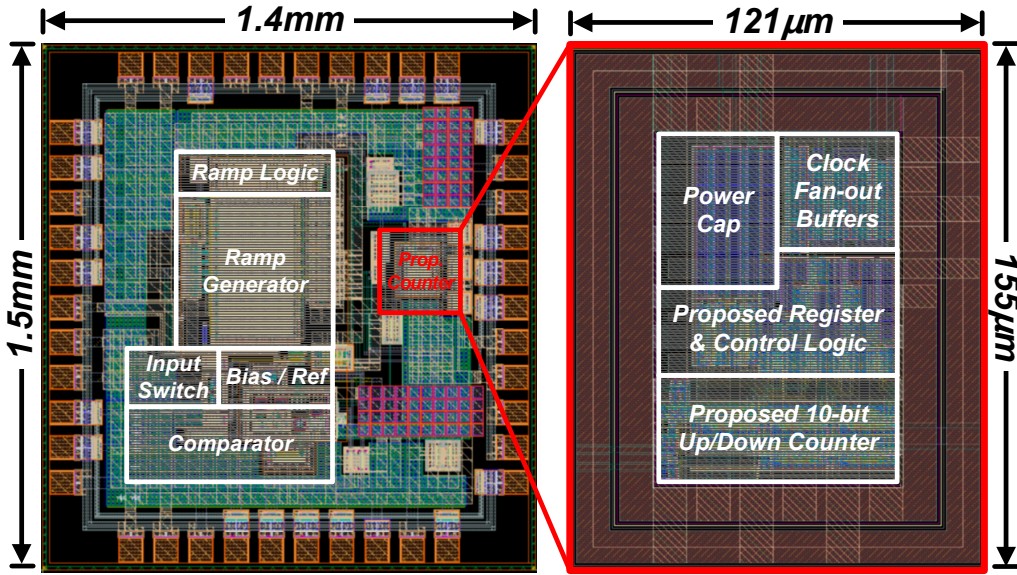

**Figure 12.** Layout of proposed SS ADC.

**Table 1.** Comparison of conventional and proposed counter.

| Parameter | | Conventional Counter | Proposed Counter and Register |
|---|---|---|---|
| Total Number of Transistors | Counter Unit Cell (T-FF) | 420 (×10 units) | 462 (×11 units) |
| | Counter Unit Cell (T-FF) | 0 | 308 (×11 units) |
| | MUX Switch | 0 | 434 (×31 units) |
| | Control Logic Gate | 18 | 18 |
| | | 438 | 1222 |
| Digital Power Consumption | | 35.51 μW | 39.42 μW |
| Area Consumption of Counter | | 121 μm × 113 μm = 0.136 mm$^2$ | 121 μm × 155 μm = 0.187 mm$^2$ |
| Unique Function | | - | Baseline Digital Calibration |

**Table 2.** Performance summary.

| Parameter | Value |
|---|---|
| Technology | 0.18 μm 1P6M CMOS Process |
| Supply Voltages | 2.8 V (Analog), 1.8 V (Digital) |
| ADC Input Range | 1.2 V |
| Power Consumption | 1.649 mW |
| ADC Resolution | 10 bits |
| SNDR | 57.56 dB @$F_{in}$ = 164.06 Hz |
| ENOB | 9.27 bits @$F_{in}$ = 164.06 Hz |
| DNL | +0.259/−0.263 |
| INL | +0.868/−0.122 |
| Sampling Rate | 28 kHz |
| Sensing Type | Resistance |
| Sensing Range | 10~120 kΩ |
| Unique Function | Baseline Digital Calibration |

## 5. Conclusions

This work introduces an SS ADC design integrated with a simple but strong baseline calibration scheme to enhance the accuracy and reliability of gas sensor readings. Various simulation results confirmed the efficacy of the proposed baseline calibration scheme, which implies that it would successfully address the challenge of baseline variability. This work offers a robust, precise, and power-efficient baseline calibration solution that meets the increasing demands for reliable gas detection in various applications.

**Author Contributions:** Conceptualization, J.-S.H. and H.-J.K.; methodology, J.-S.H. and H.-J.K.; software, J.-S.H.; validation, J.-S.H.; formal analysis, H.-J.K.; investigation, J.-S.H. and H.-J.K.; resources, J.-S.H. and H.-J.K.; data curation, J.-S.H.; writing—original draft preparation, J.-S.H. and H.-J.K.; writing—review and editing, H.-J.K.; visualization, J.-S.H.; supervision, H.-J.K.; project administration, H.-J.K.; funding acquisition, H.-J.K. All authors have read and agreed to the published version of the manuscript.

**Funding:** This study was supported by the Research Program funded by the SeoulTech (Seoul National University of Science and Technology).

**Data Availability Statement:** All data underlying the results are available as part of the present article and no additional source data are required.

**Conflicts of Interest:** The authors declare no conflicts of interest.

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
