# Peer review of "Baseline Calibration Scheme Embedded in Single-Slope ADC for Gas Sensor Applications"

_electronics, doi:10.3390/electronics13071252_

Round 1

Reviewer 1 Report

Comments and Suggestions for Authors

Reviewer 2 Report

Comments and Suggestions for Authors

This manuscript presented a single-slope analog-to digital converter with baseline calibration function to improve its accuracy and reliability of gas sensor. The design ideal is a little bit like a boxcar integrator. Clear description and detailed discussions including performance comparison have been presented. The figures and captions are also clear. The simulation results also supported its good performance. I recommend this manuscript is good and ready for publication.

Reviewer 3 Report

Comments and Suggestions for Authors

The manuscript authored by Hyeon et al. introduces a novel calibration scheme for gas sensor signals, effectively tackling the challenge of sensor baseline shifts over time and the resultant inaccuracies in sensing signals. After addressing the following issues, I recommend it for publication in Electronics.

1.     Currently, numerous gas sensors utilizing resistive changes are commercially available. Nevertheless, these sensors often suffer from short lifespans, operate within restricted temperature ranges, rely on oxygen, and face challenges operating in complex electromagnetic and harsh environmental conditions. Related to recently reported optoelectronic gas sensors, e.g. DOI: [10.1063/5.0164107, 10.1364/PRJ.492473, 10.1002/adma.202207777], does the authors’ new scheme demonstrate remarkable advances? I suggest they can discuss it.

2.     Does the proposed calibration method require a stable environment free of target gases (such as clean air) for baseline data calibration? If so, does this suggest that the method may be unsuitable for sensors designed for continuous monitoring in a target environment where they remain undisturbed for extended periods?

3.     As shown in Figure 1, the baseline is not constant, displaying significant drift over 100 seconds. Can the authors offer explanation more in details.

4.     Without altering the sensor structure, perform two-step data readings on traditional gas sensors. Utilize the data from the first reading as a reference point and calculate the difference between the second reading and the first. Can this approach achieve similar results to the method proposed in this paper?

5.     In this manuscript's figures, several areas are obscured. Please review and amend them accordingly.

Overall, I suggest a minor revision for this manuscript.

Round 2

Reviewer 1 Report

Comments and Suggestions for Authors

The author has answered all my questions and corrected any formatting errors. Agree to accept.